# Differential Sensitivity to MEK Inhibitors Highlights Distinct Entosis Mechanisms in BxPC3 and MCF7 Cells

**DOI:** 10.3390/cells14191500

**Published:** 2025-09-25

**Authors:** Paweł Tyrna, Julia Kostro, Monika Olszanecka, Piotr Szukało, Izabela Młynarczuk-Biały

**Affiliations:** 1Histology and Embryology Students’ Science Association, Department of Histology and Embryology, Faculty of Medicine, Medical University of Warsaw, Chalubinskiego 5, 02-004 Warsaw, Poland; ptyrna@cmkp.edu.pl (P.T.); julikostro00@gmail.com (J.K.); olszaneckamonika@gmail.com (M.O.); p.szukalo@nencki.edu.pl (P.S.); 2Department of Cell Biology and Immunology, Centre of Postgraduate Medical Education, Marymoncka 99/103, 01-813 Warsaw, Poland; 3Laboratory of Brain Imaging, Nencki Institute of Experimental Biology, Polish Academy of Sciences, Pasteura 3, 02-093 Warsaw, Poland; 4Department of Histology and Embryology, Faculty of Medicine, Medical University of Warsaw, Chalubinskiego 5, 02-004 Warsaw, Poland

**Keywords:** entosis, MEK, Rac1B, adhesion, cancer invasion, cell migration, stress fibers

## Abstract

Entosis is a form of cell-in-cell interaction observed in epithelial cancers, characterized by the internalization of one cell into another. This process is initiated by cell detachment, cadherin-mediated homotypic adhesion, and the formation of an entotic vacuole. Mechanistically, entosis is driven by Rho/ROCK signaling and actomyosin contractility in the invading (inner) cell, which becomes stiffer and is pulled into the softer host (outer) cell. A functional assay using differently stained cell populations allows for the assessment of pharmacological interventions on either the inner or outer cell during entosis. In this study, we investigated the impact of MEK pathway inhibition on entosis in two epithelial cancer cell lines, BxPC3 (pancreatic cancer) and MCF7 (breast cancer). BxPC3 cells, which rely on adhesion, exhibited a significant reduction in entotic index upon MEK inhibition. In contrast, MCF7 cells showed no selectivity of entosis to three different MEK inhibitors. These findings suggest cell-type-specific regulation of entosis, potentially linked to differences in protrusion formation mechanisms and upstream Ras signaling pathways previously implicated in cancer cell motility.

## 1. Introduction

Entosis is a process in which one epithelial cell enters another cell’s cytoplasm, forming a cell-in-cell (CIC) structure. Entosis was first reported by Overholtzer et al. in 2007 as a new type of cell death, because the majority of cells performing entosis were killed and digested by lysosomal enzymes [1]. Since then, our understanding of entosis and its molecular mechanisms has increased. Because the engulfed cells sometimes remain viable for extended periods of time, the Nomenclature Committee on Cell Death proposed to define entosis as the internalization process only, whereas the subsequent death of the inner entotic cell should be referred to as “entotic cell death” [2].

Entosis has been identified in several types of cancer, but its clinical consequences remain unclear. On one hand, entosis is believed to possess tumor-suppressive activity by removal of cancerous cells, e.g., aneuploid cells after aberrant mitosis [3,4]. On the other hand, entosis is known to induce aneuploidy and increase genomic instability of the outer entotic cell [5]. In general, entosis occurs in more malignant and advanced tumors, and is associated with worse prognosis [4,6].

Entosis can be initiated by several mechanisms, such as detachment from the extracellular matrix [1], prolonged mitotic division [7], or depletion of nutrients [8,9]. Regardless of the initial stimulus, in homotypic entosis one epithelial cell adheres to its neighbor via E-cadherin. This adhesion activates the Rho/ROCK pathway, which is responsible for cytoskeleton remodeling in the engulfed cell [1]. This cell forms blebs with its plasma membrane in an ezrin-dependent manner, and then enters the cytoplasm of the other cell [10]. Entosis is considered an invasive CIC structure, because the inner cell is active throughout the process. This distinguishes entosis from a morphologically similar phenomenon, cell cannibalism [11,12].

The execution of entosis is relatively well characterized, but its regulation is mostly unclear. Many factors involved in entosis regulation have been described [13], e.g., the AMP-dependent kinase (AMPK) [8], myosin-related transcription factor (MRTF) [10], Aurora A kinase [14], JNK and p38 [15], CDC42 [7], and TRAIL signaling [16]. This list is certainly incomplete, and more elements are required to understand the interplay between different regulating factors [17].

We recently published an analysis of a cohort of breast cancer patients, where we observed that the number of entoses correlates with HER2 and Ki67 expression [18]. These proteins are linked via the mitogen-activated protein kinase (MAPK) pathway: HER2 is a growth factor receptor, which lies upstream from this pathway, whereas Ki67 is a proliferation marker downstream from it [19]. In this study, we aimed to verify the involvement of the MAPK pathway, specifically MEK, in entosis.

The classical MAPK pathway consists of four proteins: RAS, RAF, MEK, and ERK. The interaction between growth factors and other mitogens with their receptors results in activation of RAS, i.e., attachment of a GTP molecule. This causes sequential activation of RAF, MEK, and ERK, all of which function as kinases [20].

MEK occurs in two isoforms, MEK1 and MEK2. The only identified substrate of MEK1/2 is ERK, which also exists in two isoforms, ERK1 and ERK2. The two isoforms of both MEK and ERK exhibit structural and functional similarities, and therefore will be collectively referred to as “MEK” and “ERK”. MEK functions as a dual specificity kinase, i.e., this enzyme phosphorylates both tyrosine and threonine residues of ERK. When phosphorylated, active ERK translocates from the cytoplasm to the nucleus to modulate gene expression profile [21].

MCF7 (from breast cancer) and BxPC3 (from pancreatic cancer) are the two human cell lines most commonly recognized in the literature as entosis-competent models. In our previous work, we described differences in the entosis scenarios of the BxPC3 and MCF7 cell lines [4]. MCF7 cells undergo entosis under matrix-deprived conditions, while BxPC3 cells exhibit spontaneous entosis even in adherent cultures [1,4,22]. Together, they provide complementary systems for studying the mechanisms and dynamics of entosis in vitro.

Both MCF7 and BxPC3 cell lines are wild-type with respect to *RAS* mutations, meaning they do not carry activating mutations in the *RAS* oncogene [23,24]. Therefore, RAS signaling does not determine the winner-loser dynamics in entosis within these cells [25]. Interestingly, both cell lines exhibit CDC42 depletion, a condition known to promote entotic competence by altering cytoskeletal tension and polarity. Although MEK kinase functions downstream of RAS and plays a well-established role in cell adhesion and cytoskeletal remodeling, its involvement in the entotic process has never been systematically investigated. This gap in knowledge is particularly relevant given MEK’s potential to influence the mechanical and adhesive properties that underlie entosis.

## 2. Materials and Methods

### 2.1. Chemicals

MEK inhibitors: cobimetinib was purchased from MedChemExpress (MedChemExpress, Monmouth Junction, NJ, USA) and u0126 was purchased from Promega (Promega, Madison, WI, USA). BI-847325, a MEK and Aurora A kinase inhibitor, was purchased from MedChemExpress (MedChemExpress, Monmouth Junction, NJ, USA). All compounds were dissolved in dimethyl sulfoxide (DMSO) at 10 mM and stored at −20 °C until use. DMSO controls were included in all experiments.

CellTracker™ dyes: CMTPX and CFDA-SE were purchased from ThermoFisher (ThermoFisher Scientific, Waltham, MA, USA) and stored as lyophilizates in −20 °C.

### 2.2. Cell Culture

BxPC3 cells (ATCC) were cultured in RPMI medium (Capricorn Scientific, Ebsdorfergrund, Germany), MCF7 cells (ATCC) were cultured in DMEM medium (Capricorn Scientific, Ebsdorfergrund, Germany). All media were supplemented with 10% Fetal Bovine Serum (Capricorn Scientific, Germany), 8 μg/mL ciprofloxacin (Polpharma SA, Starogard Gdański, Poland) and a standard antibiotic–antimycotic solution (Capricorn Scientific, Ebsdorfergrund, Germany). The cells were incubated in a humidified atmosphere enriched in 5% CO_2_ at 37 °C. The media were replenished every 2 days. Cells were passaged after reaching 60–70% confluency with the standard trypsin-EDTA solution (Biochrom AG, Berlin, Germany).

### 2.3. Cell Viability Assay

In order to select the concentrations of inhibitors for further experiments, we evaluated their toxicity against the cells. We used the PrestoBlue cell viability assay (ThermoFisher Scientific, Waltham, MA, USA), according to the manufacturer’s protocol. Briefly, cells were seeded in 96-well tissue culture plates (Corning Inc., Corning, NY, USA) at a density of 15 × 10^3^ cells per well. After 24 h, we incubated the cells with the tested compounds for another 24 h. After the incubation period, we added the PrestoBlue reagent to each well (10% *v*/*v*) and incubated at 37 °C for 40 min. Then we measured the fluorescence of each sample (excitation 544 nm/emission 610–615 nm) using the Fluostar plate reader (BMG Labtech, Ortenberg, Germany). Cell viability was calculated from the formula:$$viability\;\left(\%\right)=\frac{I-I_{ctrl-}}{I_{ctrl+}-I_{ctrl-}}\cdot 100\%$$
where *I* denotes the sample fluorescence, *I_ctrl_*_+_ denotes fluorescence of the positive control (cells treated with vehicle), and *I_ctrl−_* denotes fluorescence of the negative control (cells killed with 20% DMSO). The results were calculated as means of three repetitions.

### 2.4. Measurement of the MEK Kinase Activity

MEK kinase activity was measured by determination of concentration of its product, i.e., phosphorylated ERK (pERK). We used the DYC1018B ELISA kit (Bio-Techne, Minneapolis, MN, USA) to measure pERK contents in cells. This assay measures the total concentration of both pERK isoforms, i.e., pERK1 and pERK2.

Cells were treated with the tested compounds for 24 h. After the incubation, 1 × 10^6^ cells were lysed in 50 μL of Lysis Buffer prepared according to the protocol. The lysates were kept in −80 °C until the assay procedure. The total protein concentration was measured with bicinchoninic acid (BCA) assay (Sigma Aldrich, St. Louis, MO, USA) for normalization.

The amount of pERK in each lysate was measured according to the manufacturer’s protocol. Briefly, Nunc MaxiSorp plates (ThermoFisher Scientific, Waltham, MA, USA) were coated with pERK capture antibody and blocked with bovine serum albumin. Then, the plate was sequentially filled with the lysates, biotinylated detection antibody, streptavidin-horseradish peroxidase, and finally tetramethylbenzidine (Promega, Madison, WI, USA). The reaction was stopped with 2N H_2_SO_4_ and the optical density was determined at 450 nm (wavelength correction 540 nm) with the plate reader. The amount of pERK was calculated relative to control cells. There were two biological repetitions of each sample; they were assayed in duplicates.

### 2.5. Determination of Total Entotic Index

In order to assess the effect of each compound on the number of entoses, we generated microscope slides of cells treated with the compound. Briefly, cells were seeded on 8-well chamber slides (BD Biosciences, Franklin Lakes, MI, USA) at 3 × 10^4^ cells/well. The cells were treated with the tested compound for 24 h. After the incubation period, the cells were fixed with ice-cold 70% ethanol solution in PBS for 2 min. We removed the chamber according to the manufacturer’s protocol and stained the cells with hematoxylin and eosin. The experimental procedure is summarized in Figure 1A.

The slides were photographed under an optical microscope (Axio Vert.A1, Zeiss, Oberkochen, Germany) with the Axiocam 503 Color camera (Zeiss, Oberkochen, Germany) at 400× magnification. We generated 8 micrographs from every sample (two biological replicates, four micrographs from each) and counted the total number of cells and the number of cells involved in entosis on the micrographs. We calculated the total entotic index, defined as$$total\;entotic\;index=\frac{no.\;of\;cells\;involved\;in\;entosis}{no.\;of\;all\;cells}\cdot 100\%$$

### 2.6. Determination of Inner and Outer Entotic Indexes

To assess whether the change of total entotic index is a consequence of the compound’s activity in the inner or the outer entotic cell, we created microscope slides with co-cultures. Each co-culture is composed of two populations of the same cell line, but only one population was treated with the tested compound.

Briefly, we cultured cells on 24-well plates (Corning Inc., Corning, NY, USA) until 60–70% confluency. Then, cells were stained with fluorescent CellTracker™ dyes (ThermoFisher Scientific, Waltham, MA, USA): red CMTPX or green CFDA-SE, according to the manufacturer’s instructions. After staining, some cells were treated with the tested compound for 24 h. After the incubation, cells were seeded on 8-well chamber slides (BD Biosciences, Franklin Lakes, NJ USA) at 3 × 10^4^ cells/well. Each well contained a co-culture of green and red cells, and only one of these populations had been treated with the tested compound. The other population had been treated with vehicle.

After another 24 h, we removed the chamber and fixed cells with ice-cold 70% ethanol in PBS for 2 min. Then, cells were stained with Hoechst stain (Sigma Aldrich, MA, USA) and mounted in VectaShield^®^ medium (Vector Laboratories, Newark, CA, USA). The procedure is summarized in Figure 1B.

The slides were photographed under a fluorescent microscope (Zeiss Axio Vert.A1) at 400× magnification. Images from the red, green, and blue channels were merged with the ZEN Pro 3.6 software and their brightness was adjusted for best clarity. Some slides were scanned with a confocal microscope (LSM 900, Zeiss, Oberkochen, Germany) at 400× magnification.

We generated eight micrographs from each sample (two biological replicates, four micrographs from each). Moreover, in order to exclude potential influence of the fluorescent dyes, every inhibitor was evaluated twice (two replicates with red-stained inhibited cells and two replicates with green-stained inhibited cells). Subsequently, we counted the number of cells by their color and involvement in entosis. From each micrograph, four separate entotic indexes were calculated: the inner and outer entotic indexes for both populations. The red inner entotic index was defined as$$red\;inner\;entotic\;index=\frac{no.\;of\;red\;inner\;entotic\;cells}{no.\;of\;all\;red\;cells}\cdot 100\%$$

Other entotic indexes were defined analogously.

### 2.7. Statistical Analysis

We performed Kruskal–Wallis and Conover–Iman tests to test for differences between multiple subgroups. The significance level was set at 0.05. The data were stored in Microsoft Excel, and the statistical analysis was performed in R software ver. 4.0.2 [26]. Column charts were generated in Microsoft Excel 365, while boxplots were drawn in R software.

## 3. Results

### 3.1. Toxicity of MEK Inhibitors and Their Effect on MEK Activity

First, we determined the toxicity of the studied MEK inhibitors to include non-toxic concentrations for further experiments. We treated cells with each inhibitor at 0.1 μM, 1 μM, 10 μM, and 40 μM for 24 h, then measured cell viability with the PrestoBlue assay.

In BxPC3 cells, BI-847325 displayed the highest toxicity (IC50 5.5 μM), cobimetinib was less toxic (IC50 8.7 μM), whereas u0126 was the least toxic (IC50 > 40 μM). The results were similar in MCF7 cells, in particular: IC50 of BI-847325 was 10.7 μM, cobimetinib—14.3 μM, u0126—>40 μM. The results of the cell viability assay are provided in Figure 2A,C.

We also prepared microscope slides of cells treated with the inhibitors to test whether the inhibitors affected cell count and morphology. Based on the cell viability assay and direct observations of HE-stained cells, we selected non-toxic inhibitor concentrations for further experiments. BxPC3 cells were treated with 0.5 μM BI-847325, 5 μM cobimetinib, and 20 μM u0126. MCF7 cells were treated with 5 μM BI-847325, 2 μM cobimetinib, and 20 μM u0126.

In the next step, we evaluated how the inhibitors altered the activity of the MEK/ERK pathway. We measured the change in phosphorylated ERK (pERK) contents in cells after treatment with the tested compound. pERK concentration was assessed with an ELISA assay for pERK1 and pERK2. In addition, we have included representative Western blot images in Appendix A.

In both cell lines, we noted a significant reduction in pERK after treatment with each inhibitor. In BxPC3 cells, BI-847325 decreased the concentration of pERK 3.3 fold, cobimetinib—4.4 fold, and u0126—1.4 fold when compared to control. In MCF7 cells, we observed BI-847325 and cobimetinib reduce pERK 2.0 fold, and u0126—1.5 fold. The results of the ELISA assay are presented in Figure 2B,D.

### 3.2. The Effect of MEK Inhibitors on Total Entotic Index

The total entotic index was measured in hematoxylin- and eosin-stained cultures of cells treated with the studied inhibitors. The definition of the total entotic index is analogous to mitotic index, and it represents the percentage of entotic cells in a slide.

Entosis was present in both BxPC3 and MCF7 cells. Control cells displayed total entotic indexes of 4.0 ± 1.4% (BxPC3) and 5.4 ± 1.8% (MCF7).

In BxPC3 cells, all three MEK inhibitors decreased the total entotic index. The index at the presence of 0.5 μM BI-847325 was 2.4 ± 1.7% (*p* = 0.005); 5 μM cobimetinib reduced the index to 0.9 ± 1.8% (*p* < 0.001), whereas 20 μM u0126 set the index to 2.3 ± 1.7% (*p* = 0.006).

In MCF7 cells, two out of the three studied inhibitors significantly affected the total entotic index. The inhibitor 5 μM BI-847325 changed the index to 3.5 ± 1.6% (*p* = 0.008), while 20 μM u0126 reduced the index to 3.3 ± 2.7% (*p* = 0.004). Meanwhile, 2 μM cobimetinib changed the index to 4.2 ± 2.6%, which is not statistically significant (*p* = 0.073).

The results of total entotic index measurements are summarized in Figure 3.

### 3.3. The Effect of MEK Inhibitors on Inner and Outer Entotic Indexes

We aimed to determine whether the effect on total entotic index was mediated by inhibiting MEK in the inner or the outer entotic cell. Therefore, we created co-cultures of cells treated with an inhibitor with untreated (control) cells. The cells had been pre-stained with red or green fluorescent dyes prior to co-culturing, so as to identify them on the microscope slides. Entotic indexes were calculated separately for each population. For instance, the inner and outer entotic indexes of red cells were defined as percentages of inner and outer red entotic cells among all red cells on that slide, respectively.

In control BxPC3 cells, we observed an inner entotic index of 3.1 ± 2.3% and an outer entotic index of 2.5 ± 2.0%. All three studied inhibitors decreased the inner entotic index without changing the outer entotic index. While 0.5 μM BI-847325 changed the inner entotic index to 1.1 ± 1.2% (*p* < 0.001), the outer entotic index was 2.5 ± 2.5%. Moreover, 5 μM cobimetinib reduced the inner index to 1.8 ± 1.3% (*p* = 0.022), while the outer index stayed unchanged (2.7 ± 2%). Similarly, 20 μM u0126 set the inner index to 1.8 ± 1.5% (*p* = 0.018) without any change to the outer index (2.6 ± 1.9%). This proves that in BxPC3 cells MEK inhibitors decrease the number of entoses by acting on the inner entotic cell. The results obtained in BxPC3 cells are summarized in Figure 4A,C,E.

In control MCF7 cells, the inner and outer entotic indexes were 2.0 ± 0.9% and 1.9 ± 1.1%, respectively. We did not observe any significant changes in inner entotic indexes in cells treated with 5 μM BI-847325 (2.1 ± 1.0%), 2 μM cobimetinib (2.8 ± 1.5%), or 20 μM u0126 (1.8 ± 0.9%). With regard to the outer entotic index, BI-847325 and u0126 had no effect on this parameter (2.0 ± 1.0% and 2.4 ± 1.1%, respectively). Surprisingly, cobimetinib induced an increase in the outer entotic index (3.0 ± 1.3%, *p* < 0.001). Therefore, in MCF7 cells, we did not confirm the effect of MEK inhibitors on inner or outer entotic cells. The inner and outer entotic indexes in this cell line are presented in Figure 4B,D,F.

## 4. Discussion

Entosis, despite being a relatively obscure cellular phenomenon, has emerged as a process of clinical relevance due to its role as an independent prognostic marker in cancer [27,28]. Given that cancer remains one of the leading causes of premature mortality worldwide, any novel insight into its biology holds substantial scientific and clinical value. Because entosis lacks specific biomarkers, its detection relies on morphological assessment [5], making the process time-consuming and hindering broader investigations into its biological significance. It is still unclear what molecular or cellular conditions render cells capable of undergoing entosis, or what signals activate this process. It is well established that entosis is initiated by the detachment of a cell from the extracellular matrix, followed by the formation of adhesive interactions with neighboring cells. This process involves E-cadherin and β-catenin, and is subsequently driven by Rho/ROCK signaling and actomyosin filament formation within the internalizing (inner) cell [1]. To date, the most prominent molecular characteristic associated with entosis competence is the loss of *CDC42*, a gene crucial for cytoskeletal remodeling and adhesive interactions [4]. Beyond genetic alterations such as *CDC42* deletion, entosis competence has also been linked to pharmacological agents like nintedanib [29], activation of Aurora A kinase [14], metabolic stress induced by AMPK depletion [8,9,30], glucose starvation [8], and oncogenic *KRAS* overexpression [25,31,32].

The *RAS* oncogene, frequently hyperactivated in human cancers, is a key component of the RAS/RAF/MEK/ERK signaling cascade. The MEK-ERK cascade orchestrates key checkpoints of the cell cycle, serving as a pivotal engine of cellular proliferation and growth [20,33]. Within this pathway, MEK kinases not only regulate cell cycle progression but also contribute to cell adhesion and the formation of actomyosin fibers [20,21,33,34,35,36].

Given the established role of MEK kinases in regulating cell adhesion, their inhibition—resulting in cell rounding, retraction of cellular extensions, and detachment from the substrate—may influence the initiation of entosis. Despite this plausible link, the direct contribution of MEK activity to entotic processes has not been systematically explored. To address this gap, we conducted a comprehensive analysis of the effects of MEK inhibitors on the frequency of entotic events.

For purposes of this work, we distinguish between inner and outer entotic cell.

The inner entotic cell is the one that actively invades and becomes internalized within another living cell. It may undergo lysosomal degradation (entotic cell death), but in some cases, it can survive and even escape.

The outer entotic cell is the host that engulfs the other cell. Its ability to engulf is influenced by mechanical deformability, myosin activity, RhoA/ROCK signaling, and oncogenic mutations like *KRAS*. The outer cell may gain proliferative or metabolic advantages after digesting the inner cell [3,13,37].

Our experimental model included two entosis-competent cell lines: MCF7 and BxPC3. MCF7 cells were first described as capable of entosis by Overholtzer [1], with experiments performed under non-adherent conditions that promoted entotic structure formation in 30% of cells and lysosomal elimination of 70% of the internalized cells. Given that BxPC3 cells require adhesion for viability [4,25,34], to ensure consistent interpretation of the results, all experiments were performed under fully adherent conditions.

For inhibition of signaling, we used: u0126—an unspecific MEK inhibitor [38], which might also affect other kinases; cobimetinib—a clinically used MEK inhibitor [39,40]; and finally BI-847325—a MEK and Aurora A kinase inhibitor [41,42,43,44], to study various aspects of the MEK pathway on the frequency of entosis. In both cell lines, all inhibitors reduced the frequency of entotic structures and the reduction was statistically significant, with the exception of cobimetinib in MCF7 cells.

In order to verify which cell, the inner or the outer, requires active MEK for the execution of entosis, we conducted selective inhibition in one cell population and co-cultured it with non-inhibited homotypic cells. In BxPC3 all three inhibitors turned out to affect inner entotic cells and not for the outer ones. This is in accordance with results presented by Overholtzer in MCF7 cells for other entosis inhibitors (latrunculin, blebbistatin or RHO inhibitors) [1]. These results reinforce that inner cells drive the process of entosis in BxPC3 cells.

In contrast, all three MEK inhibitors tested failed to affect inner/outer cell preference during entosis in MCF7 cells, as neither the inner nor outer cell populations showed any significant changes in co-culture assays. This lack of effect may be attributed to less effective MEK inhibition in MCF7 cells, their lower sensitivity to these inhibitors, or fundamentally different mechanisms regulating entosis in the two cell lines. Moreover, the relative insensitivity of MCF7 cells to MEK-dependent entosis inhibition may contribute to their ability to survive under non-adherent conditions, in contrast to BxPC3 cells, which are highly dependent on adhesion [4] and exhibit MEK-dependent entotic activity. These differences may also reflect distinct sensitivities to detachment stress and the involvement of alternative Rho-GTPase signaling pathways, as previously discussed for cancer cell invasiveness by Clayton et al. [36]. Briefly, their involvement in metastatic progression have been identified: RAC1-dependent elongated migration (Figure 5), associated with extracellular matrix (ECM) degradation, and RhoA/C-dependent rounded, bleb-based mobility (Figure 6) [45,46,47].

The latter shares most similarities with entosis: the central role of RhoA and lack of ECM degradation [3,7,25]. It is possible that the differing sensitivities of MCF7 and BxPC3 cell lines are linked to variations in the regulation of cellular protrusions.

Transitioning to the rounded migration mode typically involves the suppression of RAC1 signaling, which can occur via activation of the RAC GTPase-activating protein ARHGAP22 downstream of ROCK. During elongated migration, RAC1 not only promotes protrusive activity but also inhibits the contractile forces necessary for rounded motility through its effector WAVE2, which downregulates myosin light chain (MLC) phosphorylation [46].

Importantly, cancer cells demonstrate plasticity in switching between these two motility modes. This adaptability implies that effective anti-metastatic strategies may require simultaneous targeting of both mechanisms—for instance, by combining inhibitors of ECM-degrading proteases with ROCK pathway antagonists [47]. There is no data linking RAC1 protein expression directly to entosis, but there is evidence linking this protein to cell invasiveness in both MCF7 and BxPC3 cell lines. Experiments were performed by two independent groups with overexpression, knockdown, or inhibition of this protein in these cell lines [48,49].

In MCF7 cells, under certain conditions, RAC1 can exist in the alternatively spliced form RAC1B, which is constitutively active. RAC1 is the predominant form under adherent conditions. However, overexpression of RAC1B in MCF7 cells leads to increased invasiveness (Figure 7) [48].

The opposite effect of RAC1B was shown elsewhere in BxPC3 cells [49]. Briefly, in pancreatic ductal adenocarcinoma (PDAC) cells, including BxPC3 cell line, overexpression of RAC1B has been shown to suppress TGF-β1-induced epithelial-mesenchymal transition (EMT) and cell migration. RAC1B promotes epithelial characteristics by upregulating E-cadherin and inhibiting mesenchymal markers such as Vimentin, SNAIL, and SLUG (Figure 7).

Mechanistically, RAC1B interferes with TGF-β1-induced MEK-ERK signaling, thereby protecting pancreas cancer cells from acquiring motile and invasive properties. These findings suggest that RAC1B acts as a negative regulator of EMT and may help maintain a more differentiated, less aggressive tumor phenotype in pancreatic cancer [49].

RAC1 overexpression has been shown to promote increased invasiveness, metastasis, and treatment resistance in several independent studies using the BxPC3 and MCF7 cell lines [50,51,52,53]. Therefore, targeting RAC1 may offer therapeutic benefits by limiting these aggressive behaviors. Since MEK kinase functions downstream of RAC1 in the signaling cascade and is involved in regulating cell adhesion and cytoskeletal remodeling, MEK inhibition may also reduce invasiveness by modulating RAC1 activity. Notably, in the BxPC3 pancreatic cancer cell line, MEK inhibition could potentially suppress the entosis process, which may translate into reduced invasiveness in clinical settings.

In our previous work, we postulated distinct pro-survival and lethal entotic pathways in BxPC3 and MCF7 cells, which may also be present in vivo [4,27]. In this paper, we discussed two crawling mechanisms involved in cancer cell invasiveness, highlighting the dynamic switching between Rho and RAC signaling in actin cytoskeletal plasticity and during transitions between mesenchymal and epithelial phenotypes. Importantly, it points to a gap in current knowledge regarding entosis: only indirect evidence links the Rho–RAC interplay with entosis, and it has not been directly studied in entotic models, such as BxPC3 or MCF7 cells. In this manuscript, we further emphasize a discrepancy between these cell lines, revealed by the differential impact of MEK inhibition on entotic engulfment.

Overall, entosis is still a poorly characterized phenomenon that requires extensive research, and more molecular pathways need to be explored for potential clinical applications. As characterized in previous articles [4,28], entosis can be either pro-survival or lethal to the inner cell. Entosis can have different dynamics, and a small change in signaling pathways can lead this process to different outcomes for the inner cell.

Our results also point at the difficulties in studying entosis and the need to perform experiments on multiple cell lines. Despite coherent results obtained in one cell line (BxPC3), we observed hardly any effect of MEK inhibition on entosis in co-culture assays in another cell line (MCF7). The frequency of entotic structures and their biology might vary from one cell line to another, from adherent to non-adherent conditions used in experiments.

One of the main challenges in studying entosis across multiple cell lines is that only a limited number of commercially available cell lines exhibit this phenomenon. Moreover, variations in culture conditions—such as semi-adherent environments—can significantly alter the course of entosis. Additionally, it remains unpredictable which cell types are likely to undergo entosis, making model selection and experimental design particularly difficult. However, since entosis may serve as a predictive marker in human cancers [4,5,54], studying this phenomenon could have a tangible impact on clinical outcomes for oncological patients.

## 5. Conclusions

In this study, we investigated the impact of selected MEK inhibitors on the entosis process in two entosis-competent cell lines, MCF7 and BxPC3. Inhibitor concentrations were carefully selected based on viability assays to ensure MEK activity was suppressed without significantly affecting cell survival. BI-847325 and cobimetinib, both potent and selective MEK inhibitors, demonstrated effective MEK inhibition in both cell lines. In contrast, U0126, a less selective inhibitor, targets multiple kinases, including potentially ERK5, necessitating its inclusion for comparative analysis and accurate interpretation of results.

Our findings revealed both shared and distinct features of entosis between MCF7 and BxPC3 cells, with MEK involvement emerging as a key differentiating factor. MEK inhibition led to a reduction in the overall entotic index in both cell lines. However, a more pronounced effect was observed in BxPC3 cells, where MEK inhibition completely abolished inner cell invasion, indicating a strong dependence on MEK signaling. In MCF7 cells, the inner entotic index was less affected, suggesting a more moderate role for MEK in regulating entosis. These results can be explained by the differential impact of proteins regulating actin filament formation and invasiveness between the two cell lines. More research will be necessary to verify this hypothesis.

The results reported in this paper highlight the differential contribution of MEK signaling to entosis in distinct cellular contexts and underscore the importance of cell-specific factors in the regulation of this process.

## Figures and Tables

**Figure 1 cells-14-01500-f001:**
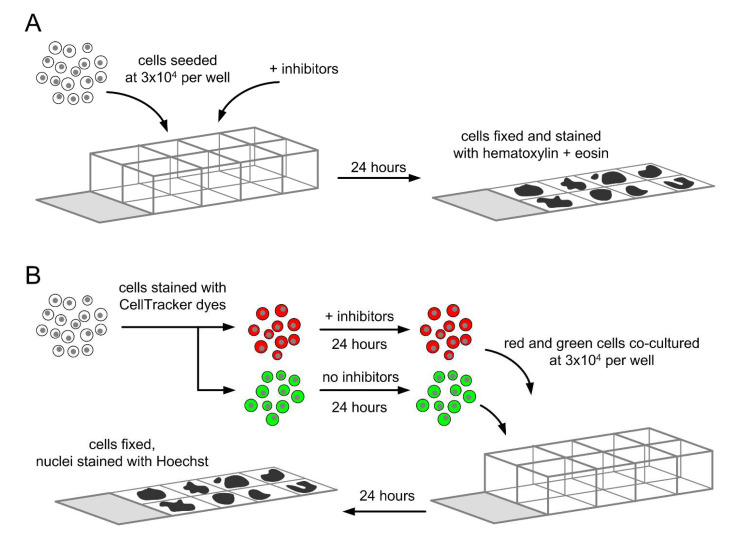
Schematic representation of experimental procedures. (**A**). Preparation of monocultures for determination of total entotic index. Cells were seeded at 3 × 10^4^ per well on 8-well chamber slides. MEK inhibitors were added to some wells. After 24 h, the cells were fixed and stained with hematoxylin and eosin, and analyzed under light microscope. (**B**). Preparation of co-cultures for determination of inner and outer entotic indexes. Cells were first stained with either red or green CellTracker dyes. Then, only one population (red in the figure) was treated with a given MEK inhibitor for 24 h. Then, red and green cells were mixed and co-cultured at 3 × 10^4^ per well on 8-well chamber slides. After 24 h, the cells were fixed, their nuclei were stained with Hoechst, and the slides were analyzed under a fluorescent or confocal microscope.

**Figure 2 cells-14-01500-f002:**
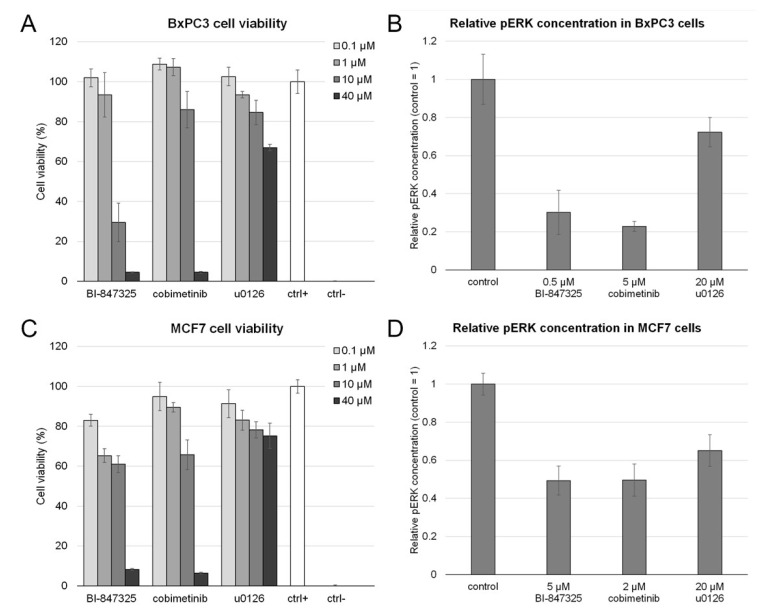
The toxicity and change in pERK concentration in cells treated with the MEK inhibitors. (**A**,**C**). Cell viability of BxPC3 and MCF7 cells measured with the PrestoBlue assay in the presence of the MEK inhibitors in varying concentrations. Positive control (ctrl+) denotes cells treated with vehicle and their viability is defined as 100%. Negative control (ctrl−) are cells killed with 20% DMSO, their viability is set as 0%. Error bars present standard deviation, *n* = 3 in each group. (**B**,**D**). Relative concentration of phosphorylated ERK (pERK) in BxPC3 and MCF7 cells after treatment with the MEK inhibitors, measured with an ELISA assay, normalized to total protein concentration. In each cell line, pERK concentration in control cells is defined as 1, so that the plot indicates the fold change in pERK concentration induced by the inhibitors. Error bars present standard deviation, *n* = 4 in each group (two biological replicates, each assayed in duplicates).

**Figure 3 cells-14-01500-f003:**
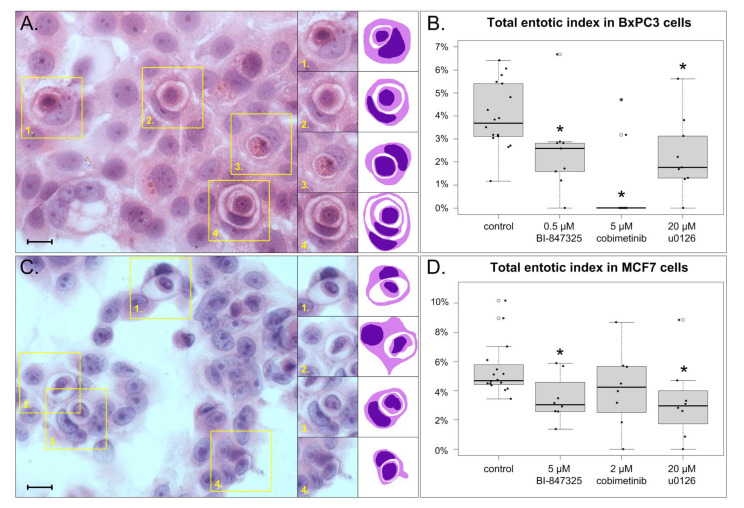
Total entotic index in cells treated with MEK inhibitors and stained with HE. (**A**). Part of a micrograph of control BxPC3 cells; yellow rectangles indicate entotic structures, redrawn for clarity on the right. (**B**). Boxplots of total entotic indexes in BxPC3 cells treated with BI-847325, cobimetinib, and u0126. Each black dot represents an index calculated from a single micrograph. (**C**). Part of a micrograph of control MCF7 cells; yellow rectangles indicate entotic structures, redrawn for clarity on the right. (**D**). Boxplots of total entotic indexes in MCF7 cells treated with BI-847325, cobimetinib, and u0126. Each black dot represents an index calculated from a single micrograph. Asterisks denote statistically significant differences from control (*p* < 0.05), *n* = 8 in every group. Scalebar 10 µm.

**Figure 4 cells-14-01500-f004:**
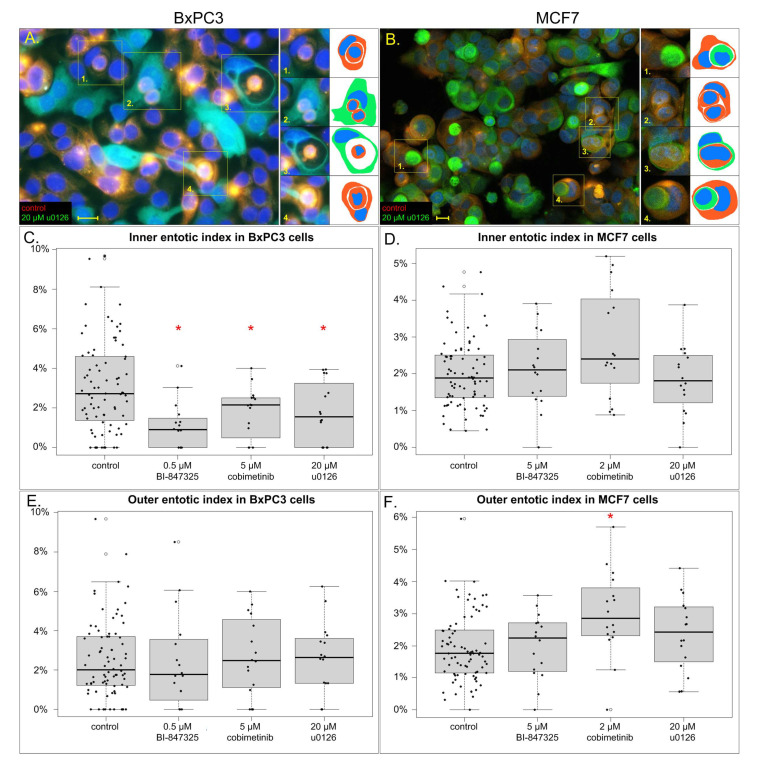
Inner and outer entotic indexes in cells treated with MEK inhibitors. (**A**). Part of a micrograph of BxPC3 co-culture, photographed with a fluorescent microscope. Green cells treated with 20 μM u0126, red cells treated with vehicle. Yellow rectangles indicate entotic structures, redrawn for clarity on the right. Note that all inner entotic cells are red. (**B**). Part of a micrograph of MCF7 co-culture, scanned with a confocal microscope. Green cells treated with 20 μM u0126, red cells treated with vehicle. Yellow rectangles indicate entotic structures, redrawn for clarity on the right. In MCF7 cells, there is no clear pattern of colors of entotic structures. (**C**–**F**) Boxplots of inner and outer entotic indexes of the tested cells treated with MEK inhibitors. Each dot represents an index calculated from a single micrograph. Asterisks denote statistically significant differences from control (*p* < 0.05), *n* = 16 for MEK-inhibited cells, *n* = 80 for control cells. Fluorescent dyes: green—CFDA-SE, red—CMPTX. Scalebar 10 µm.

**Figure 5 cells-14-01500-f005:**
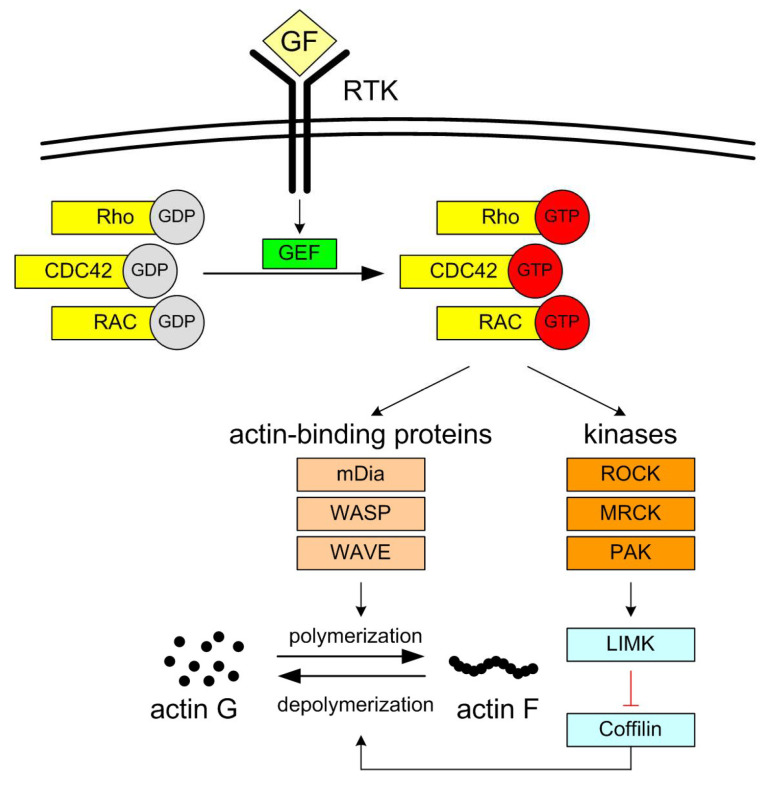
Diagram showing the role of basic cytoskeletal regulatory elements and the relationships between them. GF-Growth factors (e.g., EGF, PDGF, TGF-β) bind to receptor tyrosine kinases (RTKs) on the cell surface. Activation of RTKs triggers signaling cascades that lead to the activation of Rho GTPases (e.g., Rho, CDC42, Rac) by exchanging GDP for GTP. Activated Rho-GTP activates actin binding proteins (mDia—Protein diaphanous homolog, WASP—Wiskott–Aldrich syndrome protein, WAVE—WASP family verprolin-homologous protein) and three similar kinases: ROCK (Rho-associated protein kinase), MRCK (myotonic dystrophy kinase-related Cdc42-binding kinase), PAK (p21 Activated Kinase). The kinases activate LIMK (LIM Kinase), which inactivates Coffilin. Actin branching and polymerization is forced by actin binding proteins (mDia, WASP, WAVE) and regulated by actin depolymerization (by Coffilin). That polymerization–depolymerization balance enables cell migration by formation of filopodia, ruffles, and stress fibers. The isoforms of Rho are RhoA, RhoB, and RhoC. RAC isoforms are: RAC1, RAC2, RAC3.

**Figure 6 cells-14-01500-f006:**
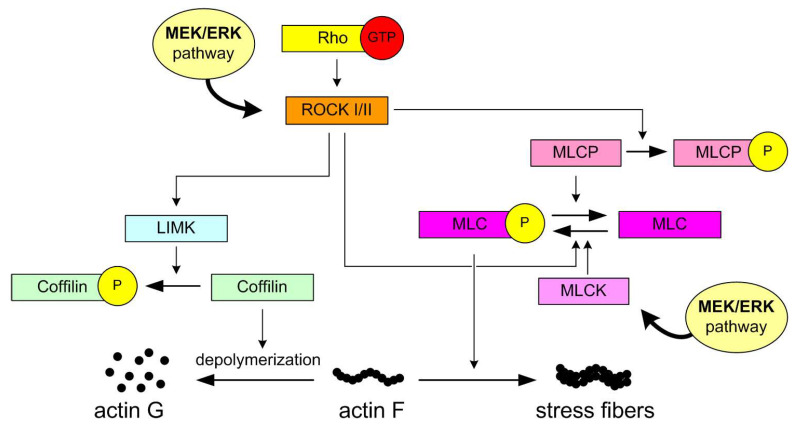
ROCK I and II (Rho-associated coiled-coil containing protein kinases) are key downstream effectors of the small GTPase RhoA and play a central role in regulating actin cytoskeleton dynamics and cellular contractility. ROCK phosphorylates and activates LIM kinase (LIMK). Activated LIMK phosphorylates cofilin, an actin-depolymerizing factor. Phosphorylated cofilin becomes inactive, leading to stabilization of actin filaments and promotion of stress fiber formation. Regulation of Myosin Light Chain (MLC): ROCK directly phosphorylates MLC, enhancing actomyosin contractility. ROCK also inhibits myosin light chain phosphatase (MLCP), which normally dephosphorylates MLC. This dual action (activation of MLC and inhibition of MLCP) results in sustained phosphorylation of MLC, increasing cellular tension and promoting stress fiber assembly. ROCK-mediated phosphorylation of MLC is calcium-independent, providing an alternative pathway for MLC activation under different signaling conditions, in contrast to calcium/calmodulin-dependent signaling.

**Figure 7 cells-14-01500-f007:**
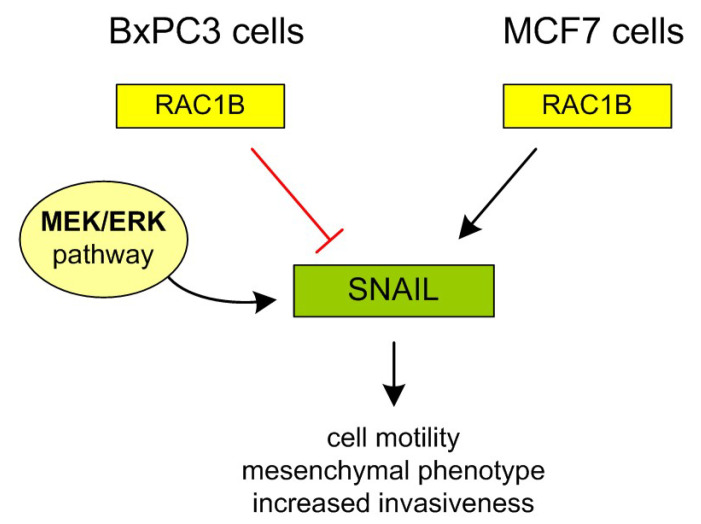
A schematic representation of the impact of RAC1B on the differential regulation of cell detachment during cancer invasion, mediated by the transcription factor SNAIL and driven by MEK/ERK signaling. In BxPC3 cells, RAC1B acts as a negative regulator of the SNAIL-dependent phenotype that promotes cell detachment, making this process more reliant on MEK activity. In contrast, in MCF7 cells, where RAC1B autonomously drives cell migration, MEK inhibition alone is insufficient to prevent detachment, as overactive RAC1B promotes both cell detachment and probably entosis in a MEK-independent manner.

## Data Availability

The data presented in this study are available on request from the corresponding author or from the first author.

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
