# Peer review of "Differential Sensitivity to MEK Inhibitors Highlights Distinct Entosis Mechanisms in BxPC3 and MCF7 Cells"

_cells, 2025, doi:10.3390/cells14191500_

Round 1
Reviewer 1 Report
Comments and Suggestions for Authors
Reviewer Report
General assessment
The manuscript by Tyrna et al. investigates the role of MEK inhibition in regulating entosis in two cancer cell lines, BxPC3 and MCF7. The study is well-motivated, given the growing recognition of entosis as a clinically relevant process in cancer biology. The experimental approach, combining viability assays, pERK quantification, and entotic index evaluation in mono- and co-cultures, is logical and technically sound. The Discussion appropriately situates the findings in the broader context of Rho/Rac signaling and cytoskeletal regulation. Overall, the manuscript addresses an underexplored aspect of cancer cell biology and provides novel insights into cell-type–specific regulation of entosis.
The manuscript is clearly written, and figures and methodology are well described. However, several issues should be addressed before the manuscript can be considered for publication.
Major Comments
- The study demonstrates differential MEK inhibitor sensitivity in two established entosis-competent lines, yet the novelty may be limited as MEK-ERK signaling in cell adhesion and motility is well documented. The authors should more explicitly clarify how their work advances beyond existing knowledge (e.g., what is fundamentally new about MEK’s role in inner versus outer entotic cells).
- While the results suggest MEK inhibition affects inner entotic cells in BxPC3 but not in MCF7, the mechanistic explanation remains speculative. The Discussion highlights Rac1B differences, but no direct experimental validation is provided. Including at least preliminary data (e.g., Rac1B expression in the two lines under tested conditions) would greatly strengthen the conclusions.
- The use of non-parametric tests (Kruskal-Wallis and Conover-Iman) is appropriate, but the manuscript should clarify the sample sizes (number of biological replicates vs. technical replicates) for each assay. At present, it is unclear whether the number of micrographs fully represents independent biological repeats.
- The Introduction and Discussion emphasize prognostic value of entosis, but the direct translational implications of MEK-dependent differences are less clear. The authors should expand on how these findings might inform therapeutic strategies (e.g., MEK inhibitors in pancreatic vs. breast cancer) or biomarker development.
- Some micrographs (Figures 3–4) lack sufficient contrast to easily identify entotic structures. Providing higher-resolution images or insets would help readers unfamiliar with entosis morphology.
Minor Comments
- The introduction is comprehensive but somewhat long. Consider condensing background sections and emphasizing knowledge gaps earlier.
- At line 83–92, the text appears to include editorial instructions (“The introduction should briefly place the study in a broad context…”). This should be deleted.
- Concentrations of inhibitors should be justified more clearly in the methods. How were final working concentrations chosen relative to IC50 values?
- Please indicate whether DMSO concentrations were controlled across all conditions.
- Provide actual n values for all quantitative analyses (not only % ± SD) in the results.
- In Figure 2, consider normalizing pERK values to total ERK to ensure that changes reflect phosphorylation rather than total protein level changes in the results.
- The section on Rho/Rac migration modes is interesting but lengthy and somewhat tangential in the discussion. Consider shortening or moving parts to Supplementary Materials.
- Clarify whether Rac1B isoform expression was directly assessed in this study, or whether conclusions are based solely on literature in the discussion.
- The manuscript is generally well written. Some sentences are overly complex and could be simplified for clarity.
- Please check reference formatting for consistency (e.g., spacing, punctuation).
Author Response
We thank the Reviewers for in depth analysis and all comments to our manuscript, which helped to improve our manuscript. We included all suggestions within the new version of our manuscript.
We did the revision short and precise – to avoid prolongation of the text of manuscript.
We included new bibliography according to comments. Song (18), Kleensang (23), Shao (24), Marcarian (37), Sun J (50), Elhansaoui (51), Xi (52), Du (53): (first author and bibliography number).
Sun now: (25) was shifted from former (30)
Shaul now: (33), was shifted from former (22)
Schenker now: (54), was shifted from former (47)
The new text is highlighted in yellow.
Moreover, we deleted redundant text originating from the template. Now is not included in the improved version.
SOME deleted text:
ERK exerts its effect on gene expression by phosphorylation of various transcription factors. ERK was proven to regulate cell cycle, survival, death, development, and other cellular processes. Several factors modulating the effect of ERK, including its specificity in a given context, have been identified to date. These include spatiotemporal distribution, interactions with scaffold proteins, crosstalk with other pathways etc.
This section may be divided by subheadings. It should provide a concise and precise description of the experimental results, their interpretation, as well as the experimental conclusions that can be drawn.
And here you can find point-by-point answer letter:
Major Comments
- The study demonstrates differential MEK inhibitor sensitivity in two established entosis-competent lines, yet the novelty may be limited as MEK-ERK signaling in cell adhesion and motility is well documented. The authors should more explicitly clarify how their work advances beyond existing knowledge (e.g., what is fundamentally new about MEK’s role in inner versus outer entotic cells). – ANSWER: we added following new text (lines: 79-93):
MCF7 (from breast cancer) and BxPC3 (from pancreatic cancer) are the two human cell lines most commonly recognized in the literature as entosis-competent models. In our previous work, we described differences in the entosis scenarios of the BxPC3 and MCF7 cell lines [4]. MCF7 cells undergo entosis under matrix-deprived conditions, while BxPC3 cells exhibit spontaneous entosis even in adherent cultures [1,4,22]. Together, they provide complementary systems for studying the mechanisms and dynamics of entosis in vitro.
Both MCF7 and BxPC3 cell lines are wild-type with respect to RAS mutations, meaning they do not carry activating mutations in the RAS oncogene [23,24]. Therefore, RAS signaling does not determine the winner-loser dynamics in entosis within these cells [25]. Interestingly, both cell lines exhibit CDC42 depletion, a condition known to promote entotic competence by altering cytoskeletal tension and polarity. Although MEK kinase functions downstream of RAS and plays a well-established role in cell adhesion and cytoskeletal remodeling, its involvement in the entotic process has never been systematically investigated. This gap in knowledge is particularly relevant given MEK’s potential to influence the mechanical and adhesive properties that underlie entosis.
- While the results suggest MEK inhibition affects inner entotic cells in BxPC3 but not in MCF7, the mechanistic explanation remains speculative. The Discussion highlights Rac1B differences, but no direct experimental validation is provided. Including at least preliminary data (e.g., Rac1B expression in the two lines under tested conditions) would greatly strengthen the conclusions. – ANSWER: by searching GeoData sets in PubMed we performed analysis of the role of Rac1 in pancreas/breast cancer biology. Thus in the discussion we included publications form other 4 independent studies on these cell lines implicating clinical correlations of Rac1 with invasiveness of these cancers, we added 4 novel citations. Novel text in Lines 437-444:
RAC1 overexpression has been shown to promote increased invasiveness, metastasis, and treatment resistance in several independent studies using the BxPC3 and MCF7 cell lines [50-53]. Therefore, targeting RAC1 may offer therapeutic benefits by limiting these aggressive behaviors. Since MEK kinase functions downstream of RAC1 in the signaling cascade and is involved in regulating cell adhesion and cytoskeletal remodeling, MEK inhibition may also reduce invasiveness by modulating RAC1 activity. Notably, in the BxPC3 pancreatic cancer cell line, MEK inhibition could potentially suppress the entosis process, which may translate into reduced invasiveness in clinical settings.
- The use of non-parametric tests (Kruskal-Wallis and Conover-Iman) is appropriate, but the manuscript should clarify the sample sizes (number of biological replicates vs. technical replicates) for each assay. At present, it is unclear whether the number of micrographs fully represents independent biological repeats. ANSWER: There were three biological replicates for viability assays, two biological replicates for pERK ELISA assay (two technical replicates for each), two biological replicates in monoculture experiments (four micrographs for each) and four biological replicates in co-culture experiments (four micrographs for each). We clarified the numbers in the methods section and in the results.
- The Introduction and Discussion emphasize prognostic value of entosis, but the direct translational implications of MEK-dependent differences are less clear. The authors should expand on how these findings might inform therapeutic strategies (e.g., MEK inhibitors in pancreatic vs. breast cancer) or biomarker development. ANSWER: In the discussion we included novel explanations: Lines 442-444: Notably, in the BxPC3 pancreatic cancer cell line, MEK inhibition could potentially suppress the entosis process, which may translate into reduced invasiveness in clinical settings.
- Some micrographs (Figures 3–4) lack sufficient contrast to easily identify entotic structures. Providing higher-resolution images or insets would help readers unfamiliar with entosis morphology. – ANSWER: The Figures were improved. For additional clarity, the indicated entotic structures were enlarged and schematically redrawn. We slightly increased the contrast on the micrographs in Figure 4. The descriptions of the Figures were updated accordingly.
Minor Comments
- The introduction is comprehensive but somewhat long. Consider condensing background sections and emphasizing knowledge gaps earlier. ANSWER: we shortened the introduction and included necessary background about selection of these cell lines and about novelty of this study Novel text in lines 79-93
- At line 83–92, the text appears to include editorial instructions (“The introduction should briefly place the study in a broad context…”). This should be deleted. ANSWER: this editorial instruction was cancelled
- Concentrations of inhibitors should be justified more clearly in the methods. How were final working concentrations chosen relative to IC50 values? ANSWER: During preliminary experiments, we performed cell viability assays and prepared hematoxylin&eosin-stained slides of cells treated with various concentrations of the tested inhibitors (0.1-40 μM). We aimed to choose concentrations which would not be toxic for the cells, both in the viability assay and in the slides (high concentrations resulted in a significantly reduced number of attached cells, which made microscopic evaluation of entosis impossible. The pre-selected concentrations were then verified to decrease pERK concentration in the ELISA assay. Thus, the concentrations satisfied both requirements: were low enough not to kill cells or make entosis evaluation impossible, but high enough to alter MEK activity.
- Please indicate whether DMSO concentrations were controlled across all conditions. ANSWER: We confirm that DMSO control was included in all experiments. This was also specified in the methods section, line: 101
- Provide actual n values for all quantitative analyses (not only % ± SD) in the results. ANSWER: We included the sample sizes in the manuscript. Each co-culture contained one MEK-inhibited population and one control population. These control populations were pooled. Additionally, we also generated control co-cultures (both populations without any inhibitor) and also included in the control group. There was no difference between the two controls.
- In Figure 2, consider normalizing pERK values to total ERK to ensure that changes reflect phosphorylation rather than total protein level changes in the results. ANSWER: We did not directly measure total ERK concentration. Samples were normalized to the same cell number and the same loading by BCA assay. Appropriate normalization was done in preliminary experiments proving that BCA protein measurement is sufficient for normalization of the method. We added this detail in the methods section, lines: 135-136
- The section on Rho/Rac migration modes is interesting but lengthy and somewhat tangential in the discussion. Consider shortening or moving parts to Supplementary Materials. ANSWER: This section was omitted, shortened with citation included that could provide the reader with appropriate information. Excluded text: Although we do not have experimental data describing crawling mechanisms in entosis, an interesting relationship was observed during cancer cell invasion. Specifically, studies in three-dimensional environments, sowed in cancer cells two distinct migration strategies, each regulated by separate Rho GTPase signaling pathways (Figure 5 and 6) [36,48,49]. The elongated mode of migration is driven by Rac1 activity, which promotes cell polarization, the formation of F-actin-enriched protrusions, and enzymatic degradation of the extracellular matrix (ECM). Conversely, signaling through RhoA/C and its downstream effector ROCK facilitates a rounded, bleb-based motility. This form of movement relies on elevated actomyosin contractility, allowing cells to navigate through pre-existing gaps and mechanically deform the ECM, often without extensive proteolytic remodeling.
- Clarify whether Rac1B isoform expression was directly assessed in this study, or whether conclusions are based solely on literature in the discussion. ANSWER: In this study we did not study RAC1B expression directly, instead we searched databases, including GEO DataSets and found 4 independent studies on RAC1 and the cell lines with clinical correlations.
- The manuscript is generally well written. Some sentences are overly complex and could be simplified for clarity. ANSWER: we improved the manuscript for clarity,
- Please check reference formatting for consistency (e.g., spacing, punctuation). ANSWER: References were reformatted by EndNote.

Reviewer 2 Report
Comments and Suggestions for Authors
In this study, the authors investigated the role of MEK pathway inhibition on entosis in BxPC3 (pancreatic cancer) and MCF7 (breast cancer).
Although the scientific subject is of interest, presented results are just preliminary based on microscopy observation, that need more appropriate experiments. In addition, the efficacy of MEK1/2 inhibitors need to be demonstrated by immunoblots.
Specific comments:
Figure 2 showed the toxicity of the studied MEK inhibitors. Among the inhibitors, UO126 20uM is not MEK specific and is known to affect othes kinase, including ERK5. Instead to use UO126, why not to use Trametinib?
Figure 2BD: Relative concentration of phosphorylated ERK (pERK) in BxPC3 and MCF7 cells after treatment with the indicated MEK inhibitors, measured with an ELISA assay, is shown. It is recommended to test the inhibitory efficacy of MEK1/2 inhibitors by immunoblots.
Figure 4 is confusing and is not convincing.
Comments on the Quality of English LanguageThe English editing should be revised.
Author Response
We thank the Reviewers for in depth analysis and all comments to our manuscript, which helped to improve our manuscript. We included all suggestions within the new version of our manuscript.
We did the revision short and precise – to avoid prolongation of the text of manuscript.
We included new bibliography according to comments. Song (18), Kleensang (23), Shao (24), Marcarian (37), Sun J (50), Elhansaoui (51), Xi (52), Du (53): (first author and bibliography number).
Sun now: (25) was shifted from former (30)
Shaul now: (33), was shifted from former (22)
Schenker now: (54), was shifted from former (47)
The new text is highlighted in yellow.
Moreover, we deleted redundant text originating from the template. Now it is not included in the improved version.
And here you can find point-by-point answer letter:
In this study, the authors investigated the role of MEK pathway inhibition on entosis in BxPC3 (pancreatic cancer) and MCF7 (breast cancer).
Although the scientific subject is of interest, presented results are just preliminary based on microscopy observation, that need more appropriate experiments. In addition, the efficacy of MEK1/2 inhibitors need to be demonstrated by immunoblots.
ANSWER: We did a grant on MEK/ERK inhibitors, the ELISA was validated by means of WB:
Specific comments:
Figure 2 showed the toxicity of the studied MEK inhibitors. Among the inhibitors, UO126 20uM is not MEK specific and is known to affect others kinase, including ERK5. Instead to use UO126, why not to use Trametinib? ANSWER: In our experiments we used U0126 together with more specific inhibitors.
1
Figure 2BD: Relative concentration of phosphorylated ERK (pERK) in BxPC3 and MCF7 cells after treatment with the indicated MEK inhibitors, measured with an ELISA assay, is shown. It is recommended to test the inhibitory efficacy of MEK1/2 inhibitors by immunoblots. ANSWER: It was checked by us in initial phase of the study showing consistent results. Moreover, the samples for the assay were prepared from an equal number of cells, which was later confirmed in total protein assessment using the BCA method. Consequently, the presented data are normalized for total protein. We would like to argue that ELISA is a reliable and validated method for protein quantification.
Figure 4 is confusing and is not convincing.
ANSWER: Figure 4 was improved and reorganized. The contrast of the micrographs was slightly increased. The indicated entotic structures were enlarged and schematically redrawn to increase clarity. We increased the font size and reduced the amount of text on the plots. Similar changes were made to Figure 3.

Round 2
Reviewer 1 Report
Comments and Suggestions for Authors
None
Author Response
We would like to thank the Reviewers for their in-depth analysis and thoughtful comments on our manuscript.
Sincerely,
Dr. Izabela Młynarczuk-Biały
Head of Cancer Cytobiology Laboratory
Reviewer 2 Report
Comments and Suggestions for Authors
The new version of the manuscript is partially improved. Although images in Fig 4 are fine in the new version, the raised question regarding Fig 2BD need to be answered. In particolar, the authors should provide qualitative immunoblot showing pERK1/2 and ERK1/2 total protein expression,. The quantification should be done using the pERK1-2/ERK1-2 ratio. In addition, the off-target effects of UO126 should be tested (for instance on ERK5 activation).
Author Response
We thank the Reviewer for their valuable comments and suggestions. Below we address the specific concerns raised:
- Figure 2BD – pERK1/2 and ERK1/2 Expression:
In the supplementary figure (Supplementary Figure S1 and S2), we now provide representative Western blot data showing both phosphorylated ERK1/2 (pERK1/2) and total ERK1/2 protein levels. These samples are the same as those used for the ELISA-based quantification of pERK1/2 presented in the main manuscript. Since the results obtained by ELISA are consistent and reproducible across experiments, we have retained the ELISA data in the main figure to ensure clarity and quantitative robustness.
The Western blot data are included as qualitative confirmation and to address the Reviewer’s request.
We included in lines 226-227 following text: In addition, we have included representative Western blot images in Supplementary Figures S1 and S2.
- Off-target Effects of UO126 – ERK5 Activation:
Regarding the Reviewer’s comment on the specificity of U0126:
We appreciate the Reviewer’s concern regarding the potential off-target effects of U0126. As noted in line 347-348 of discussion in the revised manuscript, U0126 is a non-specific MEK inhibitor. To address this limitation, we have added a dedicated paragraph in the Discussion section highlighting the constraints of using U0126 in our study. Specifically, we acknowledge that we did not investigate ERK5 or other kinases that may potentially be affected by U0126, such as MEK5 or other MAPK pathway components.
In lines 478-480 we included following text:
In contrast, U0126, a less selective inhibitor, targets multiple kinases, including potentially ERK5, necessitating its inclusion for comparative analysis and accurate interpretation of results.
To mitigate this issue and strengthen our conclusions, we have included experiments using more selective MEK inhibitors, such as cobimetinib and BI-847325, which offer improved specificity.
Indeed, we performed Western blot analyses during the initial phase of our study. The blots are now included in the supplementary materials.
Currently, our laboratory is undergoing renovations, which unfortunately makes it impossible to conduct additional experiments, including those assessing ERK5 expression.

Round 3
Reviewer 2 Report
Comments and Suggestions for Authors
All the raised questions have been addressed and the new version of the manuscript is improved.